# How Did It Get So Late So Soon? The Effects of Time Management Knowledge and Practice on Students' Time Management Skills and Academic Performance



Sebastian Trentepohl [1,*], Julia Waldeyer [1], Jens Fleischer [2], Julian Roelle [1], Detlev Leutner [3] and Joachim Wirth [2]

1   Department of Educational Psychology, Ruhr University Bochum, 44801 Bochum, Germany;
    julia.waldeyer@ruhr-uni-bochum.de (J.W.); julian.roelle@ruhr-uni-bochum.de (J.R.)
2   Department of Research on Learning and Instruction, Ruhr University Bochum, 44801 Bochum, Germany;
    jens.fleischer@ruhr-uni-bochum.de (J.F.); joachim.wirth@ruhr-uni-bochum.de (J.W.)
3   Department of Instructional Psychology, University of Duisburg-Essen, 45141 Essen, Germany;
    detlev.leutner@uni-due.de
*   Correspondence: sebastian.trentepohl@ruhr-uni-bochum.de

**Abstract:** Time management is regarded as an important prerequisite for effective and efficient learning in higher education. However, university students' time management frequently proves to be deficient, especially with freshman students, who can therefore benefit from appropriate time management interventions. The aim of this study was to compare the effects of an intervention focused on imparting time management knowledge with those of an intervention focused on time management practice. We conducted an experiment with $N = 118$ university students who took part in a course over the duration of one semester. Participants with a time management deficit at the beginning of the semester ($n = 88$) were randomly assigned to one of three experimental conditions: (a) time management knowledge, (b) time management practice, (c) control group. Exam scores at the end of the semester were considered as an indicator of participants' academic performance. The results showed significant time management improvements for both time management intervention groups, but the time management practice group appeared superior. Academic performance was better in the time management practice group also, although the results were inconsistent. The effect of time management practice on academic performance was mediated by students' time management skills.

**Keywords:** time management; self-regulated learning; academic performance; higher education

## 1. Introduction

Creating a schedule and complying to it can be a self-regulatory challenge, especially where it is meant to organize tedious yet important tasks. Such organizing behavior is a major challenge for freshman students when transitioning from school to university, as these institutions differ significantly in terms of their learning requirements [1–5]. Transitioning from school to university therefore confronts freshman students with previously unknown challenges without having developed adequate coping strategies yet, which can lead to reduced well-being, low performance, behavioral risks, and early dropout [6–11]. Besides non-university obligations, these challenges particularly concern keeping up with the university curriculum and necessitate that freshman students develop self-regulated learning routines early on [4,12,13]. In this context, effective time management is crucial for freshman students to overcome the diverse challenges of their entry into university study and in support of first-to-second year retention.

Time management is considered an integral component of self-regulated learning [4,14] and is correlated with higher academic performance [15,16]. Unfortunately, freshman students' time management skills often tend to be deficient [5,17], indicating a need for

appropriate interventions. Several intervention approaches have been developed over past years [16,18], but evidence on the effectiveness of time management interventions in supporting students' academic performance is still scarce, with the findings being mixed [16,19,20]. From the available evidence, it can be seen that different types of time management intervention show different effects on time management skills and performance. Interventions that focus on imparting knowledge about time management strategies partially lead to improvements in time management skills but not on performance measures [21,22], whereas interventions including practice in the actual use of time management strategies lead more consistently to improvements in students' time management skills and also show effects on academic performance [23,24]. These findings appear reasonable in view of the series of related knowledge and behavior components required to influence learning outcomes, as proposed by the theoretical framework of self-regulated learning [14,25].

Combined with evidence showing that having strategy knowledge does not imply the successful use of the respective learning strategies [26–28], it can be questioned whether providing students with knowledge about time management strategies, as is often implemented as a part of freshman courses aiming to support performance and retention [12,29], can be sufficient to foster students' time management skills to the extent that it positively affects their academic performance. It can be assumed that students would benefit rather more from practice in the proper use of time management strategies to foster learning performance, and from self-reflection on their strategy use to optimize future learning processes in a self-regulated manner. However, the effects of both types of time management intervention are as yet difficult to compare, since relevant intervention studies often differ in their operationalizations of time management, intervention duration, and context [16,30]. Accordingly, the aim of this study was to investigate how freshman students' time management skills can be promoted through interventions in a way that actually helps them to improve their academic performance. Therefore, we provided an intervention focusing on time management knowledge, as well as an intervention focusing on practical exercises on the use of time management strategies for freshman students in a real university context over the course of one semester, and investigated their effects on students' time management skills and academic performance at the end of the semester.

## 1.1. Self-Regulated Learning and Time Management

Self-regulated learning is regarded as a self-directed process in which learners plan, monitor, and evaluate their use of learning strategies to achieve specific learning goals [31,32]. Although models of the process of self-regulated learning differ in nomenclature and segmentation, they commonly include a cycle of multiple distinguishable phases (for an overview, see [25]). These phases conventionally refer to a series of related *knowledge* and *behavior* components [14,33,34]. Accordingly, the phases of self-regulated learning can be referred to as the *forethought*, *performance*, and *self-reflection* phases, each of which comprises sets of related sub-processes [14,30]. During this process, learners need to evaluate the task at hand, as well as their own knowledge and skills, to plan their learning activities appropriately (*forethought phase*), use appropriate learning strategies to enact their learning plan and monitor their progress during task performance (*performance phase*), and reflect on recent strategy use to adapt their existing knowledge of relevant learning strategies and thereby optimize future learning behavior (*self-reflection phase*).

An important component of effective self-regulated learning is efficient time management. The use of time management strategies is associated with the use of other types of strategic self-regulatory behaviors that support learning activities, including cognitive, metacognitive, and other resource-management learning strategies [17,35–42]. Time management is an integral component of prominent theories on self-regulated learning, where it is considered either explicitly as a behavior that learners can actively control to self-regulate their learning activities [4,14], or implicitly as part of other regulatory processes, such as volitional strategies or goal setting and planning [43,44].

Time management can be defined as clusters of behavioral skills that are beneficial in the organization of study and course load, and that help learners to facilitate their productivity to achieve their learning goals [35,45,46]. These skills include assessment behaviors aiming at awareness of time use, planning behaviors aiming at selecting and setting up realistic goals, and monitoring behaviors aiming at the observation of time use while performing activities and reflecting on previous organizing behavior. The processes that are subject to time management refer to these forethought, performance, and self-reflection phases, as outlined in the self-regulated learning framework (for a detailed review, see [30]). In the forethought phase, learners need to activate knowledge regarding time management strategies to analyze the task at hand by gathering information regarding the estimated time needed for task completion, as well as any relevant deadlines, and then plan their learning activities by setting goals and priorities within the given timeframe to establish time-related standards for progress or success. In the performance phase, learners need to initiate the use of their strategic plans, consider the planned time and duration of relevant learning activities, and monitor compliance with their learning schedule. In the self-reflection phase, learners reflect on their learning activities by evaluating time-related experiences and outcomes, such as the chronology of task completion, their actual time investment, and whether deadlines have been met, to adapt their time management strategy knowledge and optimize their prospective use of time management strategies.

### 1.2. Time Management in Higher Education

The university learning environment poses diverse challenges to students' organizing behavior. Efficient time management is an essential tool to cope with these challenges without falling behind in the curriculum and eventually dropping out. It helps students to understand the effort that is required for effective learning and enables them to structure their learning activities and develop appropriate study habits [5,47,48]. Accordingly, empirical studies frequently demonstrate a significant correlation between students' time management and academic performance [15,16,35,38,39,49–54], as well as well-being factors such as lowered stress and anxiety [15,55–58].

Unfortunately, freshman students' time management skills tend to be deficient at the beginning of their academic career. They underestimate the time required to study successfully [17,59] and report problems in regulating study time and class attendance alongside non-university obligations [3,5,60]. They spend a considerable amount of time on activities that are not conducive to their academic performance or that distract them from learning activities, such as social networking or watching TV [61–64]. Overall, university students appear to be especially prone to procrastination [65–67] and report related self-handicapping behaviors even during class attendance [68–70]. It can be summarized that time management is a common problem, especially among freshman students, and that time management interventions can be an important tool to facilitate the challenging study entry phase, to foster performance and reduce dropout. Indeed, there is some evidence supporting the effectiveness of interventions in enhancing time management skills and performance [18,20,23,24,55,71–75], although findings regarding the effects on performance variables are mixed [15,16,21,22,52,76–78].

### 1.3. Intervention Approaches and Related Issues

There are different possible explanations for the inconsistent state of research on the effectiveness of time management interventions that aim to foster time management skills and academic performance. First of all, relevant studies tend to focus on workplace settings rather than academic contexts, and research on the effectiveness of time management interventions in fostering academic performance is comparatively scarce [16,19,20]. Moreover, existing time management interventions for freshman students do not always allow conclusions to be drawn about their effects on academic performance. For example, time management instruction based on knowledge transfer is quite common as a part of freshman courses aiming to support performance and retention [12,29,74,79]. Corresponding

programs have indeed been found to improve academic retention and graduation rates, but with time management here being one part of more general study orientation or learning strategy courses, these programs usually do not provide reliable evidence about the isolated effects of time management instruction on academic performance.

Another important issue with time management intervention studies is a lack of consistency in the conceptual understanding and measurement of time management [16]. Although time management is considered as an important component of self-regulated learning, its operationalizations often lack reference to the successive phase structure provided by process models of self-regulated learning [14,30]. Given this theoretical framework, effective time management interventions should address processes relevant to the forethought, performance, and self-reflection phases of the learning cycle, to provide students with the strategy knowledge required for the situationally appropriate use of time management strategies, and to enable them to optimize their time management behaviors over time via practice and self-reflection. This is supported by evidence showing that students' individual self-regulation deficits differ in terms of the phase of the self-regulated learning process in which they occur, and the learning strategies they affect [4,14,28,34,80]. For example, some students might lack basic knowledge (*forethought phase*) regarding a specific learning strategy, whereas other students have the required basic knowledge, but fail to use the strategy successfully (*performance phase*) to overcome learning difficulties [13,27,81]. In this context, several studies have found that students tend to have sufficient (declarative) knowledge about learning strategies, but still do not use the corresponding strategies successfully in relevant learning situations [26,27,82,83]. These findings indicate that students cannot necessarily transfer available strategy knowledge into successful strategy use on their own, which would be crucial for time management interventions focusing on imparting time management knowledge.

Accordingly, imparting time management knowledge should be a useful way to promote students' time management knowledge, but might not provide them with the skills required to develop the self-regulated time management behaviors they need to improve performance. This is reflected by results from time management intervention studies aiming to improve time management behaviors and performance via time management instruction. Macan [21] (Study 1), for example, tested the effects of an intervention that provided information on central time management behaviors such as goal setting, prioritizing, scheduling, and planning. The instruction consisted of a single session that lasted half a day and used multiple methods to teach time management strategies, including lectures, group discussions, and films. Apart from a small increase in the self-reported use of goal setting, the time management instruction had no significant effects on participants' time management behaviors and performance ratings. In a similar study where the time management instruction lasted two consecutive sessions, Macan [77] found no significant differences between a time management intervention group and a control group for post-intervention self-assessments of time management behaviors, and performance ratings were even higher for the control group than the intervention group. Moreover, Lincoln and colleagues [76] provided participants with a self-directed training package that included comprehensive information on how to improve time management skills that participants should use to improve their time management over the course of five weeks. The results showed no significant improvements in participants' time management skills. Finally, Häfner and Stock [22] carried out an experiment that entailed providing a very elaborate one-day time management instruction, supplemented by small cards with guidelines and a training booklet that afterwards should help participants to consolidate and improve the strategies learned. Participants' self-reported use of time management behaviors had significantly increased six weeks after the intervention, but there were no significant effects on performance indicators. These findings show that imparting time management knowledge can help to improve participants' self-assessments of their time management behavior. However, given the absence of performance improvements, it can be questioned whether the time management knowledge provided actually improved participants' time

management behavior, or rather their corresponding self-perceptions, on which the used self-assessments tend to rely [84–86].

In any case, time management knowledge alone seems not to imply the successful use of time management strategies, and a stronger focus on time management practice with sufficient time to foster time management routines may be important for enabling students to develop effective time management behaviors and thereby improve academic performance. Indeed, there are few studies with intervention designs that considered processes relevant to all three phases of the self-regulated learning process in training time management, which fairly consistently report improvements in learning behavior and academic performance [23,24,71]. Apart from introductions to time management strategies (*forethought phase*), students here particularly had the opportunity to deliberately practice the strategies taught on a regular basis over a longer period of time (*performance phase*). This practice was supported by learning diaries [71] or online learning systems [24], which made students use the time management strategies taught, and helped them to structure and monitor their learning activities. Furthermore, practice was supplemented by self-evaluations of learning progress (*reflection phase*), to help students optimize their strategy use over time. These findings indicate that encouraging students to practice time management and giving them sufficient time to optimize and consolidate efficient time management behaviors might be essential to effective time management interventions. However, these findings do not provide information on the specific effects that the practice focus of these interventions offers, in comparison to interventions focusing on time management knowledge, as the studies either included time management as a part of more global interventions on self-regulated learning, or did not include separate time management intervention groups so as to get information on the differing effects of time management knowledge and time management practice.

*1.4. The Present Study*

The aim of the present study was to find out how students' time management can optimally be promoted. The current state of research indicates that interventions based on imparting time management knowledge have the potential to improve students' self-reported time management, but tend not to improve their academic performance. Time management interventions focusing on time management practice, on the other hand, tend to improve students' time management and also have the potential to improve their academic performance. In both cases, intervention duration appeared to be a relevant factor in skill development, with intervention durations ranging from one single session to regular sessions over the course of several months. To test the effects of both types of intervention on students' time management and academic performance, we randomly assigned freshman students who showed a time management deficit to a time management knowledge group, a time management practice group, and a control group. All groups received a comprehensive introduction session followed by weekly exercises over the course of one semester, to assure sufficient intervention duration for skill development. Furthermore, students faced their first important exams at the end of their first university semester, and thus it appeared particularly important that they should have overcome possible learning deficits by this time, so that we aimed at this point in time for the final assessment of students' time management skills. Given the long duration between the initial and final data collection, we added another point of measurement in the middle of the semester, in order to obtain further information about the development of students' time management skills during the intervention, and the time required for the interventions to be effective.

The time management knowledge intervention focused on time management processes relevant in the forethought phase, helping students to deepen and consolidate their knowledge about time management strategies. The time management practice intervention focused on processes relevant in the performance and self-reflection phases, guiding students to create and comply with learning schedules and to evaluate their plans and plan compliance regularly. Dependent variables were students' time management skills

and academic performance at the end of the semester. The assessment of students' time management skills included a test of their time management knowledge and their confidence in using time management strategies in specific study situations. Students' academic performance was assessed on the basis of their exam scores at the end of the semester.

Improvements in students' time management should be reflected in their academic performance, as students' time management has been found to be a significant predictor of their academic performance [53]. However, according to process models of self-regulated learning, all three phases should be relevant in improving students' time management to an extent that also affects their academic performance [14,30]. Given the evidence showing that sufficient strategy knowledge does not necessarily lead to strategy use [26,27], it was expected that imparting time management knowledge would have a positive effect on students' time management skills, but would not significantly improve their academic performance. Students in the time management practice group, on the other hand, were expected to be well prepared to use the time management strategies taught and to autonomously optimize their time management strategy use over time via self-reflection, allowing them to improve their learning routines to an extent that would positively affect their academic performance. In summary, the present study served to test the following hypotheses:

**Hypothesis 1.** *Both the intervention that focuses on time management knowledge and that which focuses on time management practice foster the improvement of students' time management skills.*

**Hypothesis 2.** *Only the intervention focusing on time management practice but not the intervention focusing on time management knowledge fosters students' academic performance.*

## 2. Materials and Methods

### 2.1. Sample and Design

The sample consisted of $N$ = 118 first-semester university students from the domain of civil engineering (62.7% male, $M_{age}$ = 21.0 years, $SD_{age}$ = 5.3 years). Students voluntarily participated in a course that was part of the elective subjects scheduled in their degree program's curriculum. All data were collected during course sessions. Participants received 100 € as compensation for completing the study, and three credit points for successfully attending the course. In accordance with German legislation, institutional review board approval is not required for this type of study. This study complies with human subject guidelines of national research committees, as well as the APA Ethics Code Standards.

The study followed an experimental 3 (time management knowledge group − time management practice group − control group) × 3 (pretest − mid-test − post-test) mixed design. All groups received interventions of the same duration and with comparable structure of exercises, but with different content. The design served to compare changes in participants' time management between the different intervention groups over the course of one semester.

### 2.2. Time Management Intervention Programs

Participants in the intervention groups were trained either with a focus on time management knowledge (time management knowledge group) or on time management strategy use (time management practice group). The control group received an introduction to academic writing that was not related to time management. All interventions included a two-hour introductory session at the beginning of the semester, followed by weekly 15-min exercises over the course of the semester. The introductory session took place during a regular course meeting and the weekly exercises could be submitted using an online tool that was part of the course within which the time management intervention was housed. The submissions were checked regularly and students received personalized emails to remind them of their participation if they had not yet made a submission at the beginning of a week.

The introduction session for the *time management knowledge group* provided information about the relevance of an efficient time management for academic performance, and a detailed lecture on time management strategies including time assessment, prioritizing, goal setting, planning, monitoring time use, and reflecting on past plans [21,47]. The learning objective of this introduction was to provide participants with the knowledge required to organize relevant goals into realistic plans and to optimize these plans over time. The weekly exercises in the time management knowledge group aimed to repeat central aspects of the provided time management knowledge via short recall tasks, and to consolidate this knowledge via open-ended questions aiming at comprehension and transfer.

The introduction session for the *time management practice group* basically conveyed the same time management strategies as for the time management knowledge group. However, given the evidence showing that strategy knowledge tends to be less of a problem for students [27], the theoretical input was reduced to a very brief introduction on central time management strategies, with a strong focus instead on practical exercises in the use of the respective strategies. The practical exercises aimed at time management behaviors supporting the efficient use of time, such as setting goals, planning tasks, and prioritizing [35,87], with a focus on students' planning behavior, as this was found previously to be the most relevant time management behavior in relation to learning outcomes [16]. The exercises included creating a schedule for the following week based on the previously identified criteria for efficient time management and the use of implementation intentions [88–90] to support goal attainment. The learning objective of this introduction session was to enable students to create realistic plans and to gain initial experiences with their implementation by actively planning their own learning efforts and monitoring their planning behavior. For this purpose, each student created a personal learning schedule for the first week. Then, their schedules were discussed exemplarily in plenary with regard to central quality aspects, such as realism and specificity of the set learning goals. The weekly exercises for the time management practice group included creating a schedule with specific times and learning goals for the upcoming week, and open questions asking about compliance issues with their plan of the past week and opportunities for improvement, to help students reflect on their planning activities.

The introduction session for the *control group* provided a lesson on academic writing in order to offer a comparable training duration and simultaneously to teach students something meaningful for their studies. The weekly exercises for this group included several related tasks, such as literature research or citing according to a specific manuscript manual. The intervention for this group did not include any information on time management or self-regulated learning in general.

*2.3. Measures*

2.3.1. Time Management

Participants' time management skills were assessed using the time management subscale of the Resource-Management Inventory (ReMI) [83]. While self-assessments of strategy use are common in established inventories for assessing time management directly (e.g., TMQ [35]) or as a part of self-regulated learning (e.g., MSQL [91]), a corresponding test on the underlying time management strategy knowledge is usually not included in self-assessment instruments. Moreover, established inventories often lack context-sensitive items, which would be required to choose situationally appropriate strategies based on available strategy knowledge [43,92]. To address these issues, we decided to use a situational judgement instrument to assess students' time management skills that included a test of time management strategy knowledge. The ReMI combines a test of participants' time management strategy knowledge with a subsequent self-assessment of their ability to use the respective time management strategies in relevant learning situations.

The ReMI time management subscale comprised seven items. Each item demonstrated a learning-related problem situation and offered five different learning strategies to solve the problem. Participants had to choose one correct strategy for each situation. The time

management knowledge test was followed by a self-assessment of students' belief as to whether they would actually use the described strategies when confronted with corresponding learning problems in real life, and whether they would expect themselves to be successful doing so. Presuming that an unknown time management strategy cannot be used successfully, students' self-assessments of potential success when using the described time management strategies were only considered if they had answered the preceding time management knowledge test correctly. The resulting scores ranged from 0 to 2, with 0 indicating a lack of time management strategy knowledge, 1 indicating sufficient time management strategy knowledge but a deficit in using it, and 2 indicating sufficient time management strategy knowledge, as well as the ability to use it successfully in relevant learning situations. Item order varied across participants and time, in order to control for sequence effects. The internal consistency was acceptable; Cronbach's $\alpha = 0.69$.

### 2.3.2. Academic Performance

Participants' academic performance was assessed on the basis of an exam on the content of the course (i.e., multimedia learning) at the end of the semester. The exam included 30 multiple-choice single-selection questions, each providing one correct answer and three distractors. Each correct answer was worth one point, resulting in a maximum score of 30 points. The internal consistency was satisfactory; Cronbach's $\alpha = 0.82$.

### 2.3.3. Control Variables

Several additional inventories were used to control for the comparability of the three intervention groups. *Demographic data* included questions regarding participants' sex, age, and high-school grade point average (HSGPA). *Conscientiousness* was assessed using the corresponding subscale of the German version of the NEO Personality Inventory (NEO-PI-R) [93]. Students indicated the degree to which different conscientiousness-associated statements were descriptive of their personality on a scale ranging from 1 (*strongly disagree*) to 5 (*strongly agree*). *Trait procrastination* was assessed using the German version of the Aitken-Procrastination Scale (APS-d) [94]. The items described different types of procrastination behavior, and students indicated the extent to which they were prone to these behaviors on a scale from 1 (*strongly disagree*) to 5 (*strongly agree*). Finally, the *Time Management Questionnaire* (TMQ) [35] was included as a convergent self-report measure for students' time management, alongside the corresponding ReMI subscale. The items described different time management behaviors, and students indicated the extent to which the statements applied to them on a scale from 1 (*strongly disagree*) to 5 (*strongly agree*). The internal consistency of all included control measures was satisfactory; $0.75 \leq$ Cronbach's $\alpha \leq 0.89$.

### 2.4. Procedure

During the first course session, participants were instructed on the study, gave their written consent for participation, and completed questionnaires on demographic data and time management. They were then randomly assigned to one of three different intervention groups if they showed an initial time management deficit ($n = 88$), or were excluded from the data analysis if not ($n = 30$), since these students could not be expected to benefit from a time management intervention and would not be comparable due to their better starting condition. The groups were separated for the second course session, where they received their respective introduction and were instructed on the weekly exercises they had to submit over the course of the semester. From then on, all groups equally attended the course, in which content on multimedia learning was provided via an online-learning environment that was made accessible until the end of the semester. Participants' time management was assessed again in the middle and at the end of the semester. The course ended with an exam on multimedia learning that served as an indicator of participants' academic performance. All participants were given access to the time management instruction materials at the end of the data collection. A summary of the procedure is shown in Figure 1.

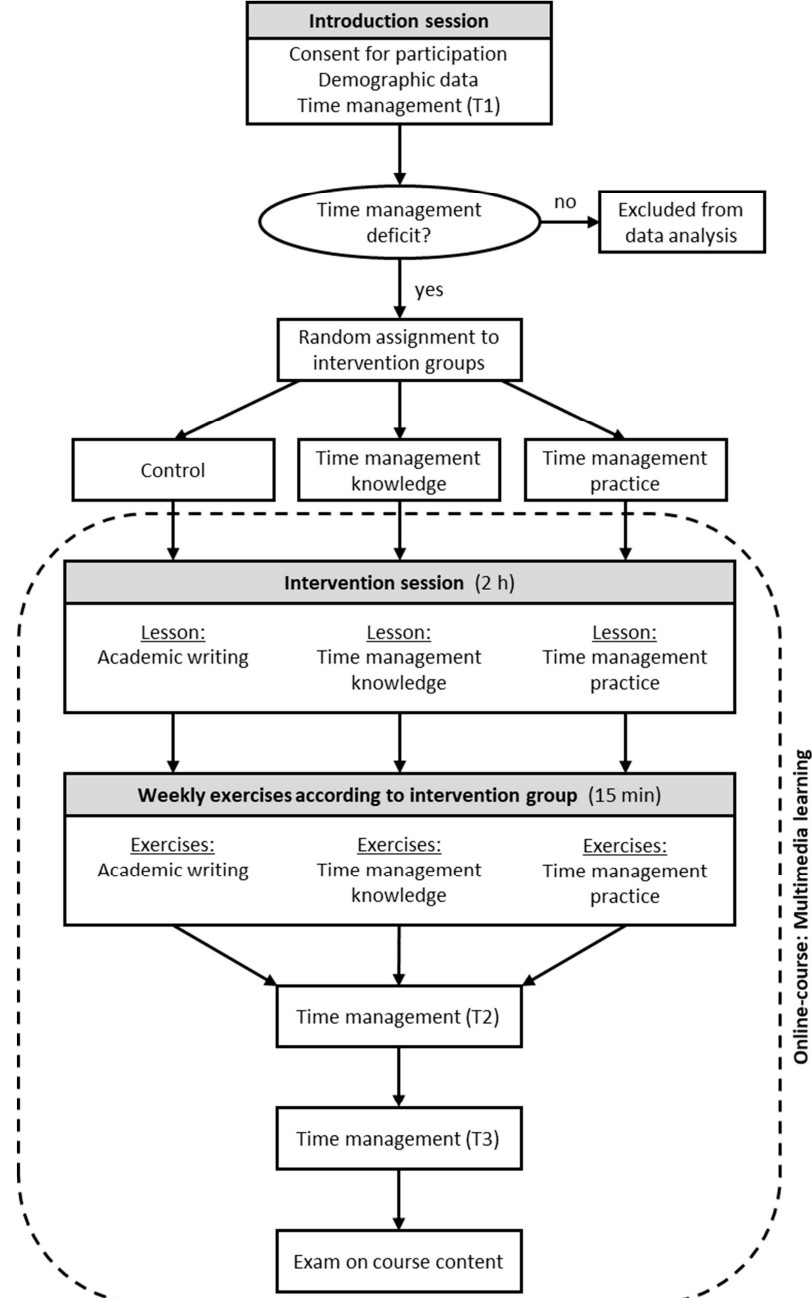

**Figure 1.** Summary of the procedure of the study, including data collection and interventions.

### 3. Results

*3.1. Preliminary Analyses*

In the first step, we tested whether the random assignment resulted in comparable groups by conducting a one-way multivariate analysis of variance (MANOVA) with group as factor and age, HSGPA, conscientiousness, trait procrastination, and time management (i.e., ReMI and TMQ) at semester start as dependent variables, and a chi-Square test for sex. There was no significant effect of group in both the MANOVA, $F(12, 164) = 0.31$, $p = 0.988$, Wilk's $\Lambda = 0.96$, $\eta_p^2 = 0.02$ (separate one-way ANOVAs: all $p = 0.301$–$0.776$), and the chi-Square test, $\chi^2(2) = 0.42$, $p = 0.810$, Cramér's $V = 0.7$, indicated that the random assignment resulted in comparable groups (see Table 1 for group means and standard deviations).

**Table 1.** Group means (*M*) and standard deviations (*SD*) of all variables in the study.

|  | Control (*n* = 26) | | Knowledge (*n* = 32) | | Practice (*n* = 30) | |
|---|---|---|---|---|---|---|
|  | *M* | *SD* | *M* | *SD* | *M* | *SD* |
| Sex (% female) | 36 |  | 39 |  | 27 |  |
| Age (years) | 20.50 | 1.84 | 21.72 | 9.01 | 20.20 | 2.34 |
| HSGPA | 2.52 | 0.70 | 2.53 | 0.67 | 2.46 | 0.61 |
| Trait procrastination | 2.50 | 0.46 | 2.47 | 0.61 | 2.40 | 0.51 |
| Conscientiousness | 3.49 | 0.45 | 3.57 | 0.63 | 3.52 | 0.42 |
| TMQ: time management | 3.04 | 0.31 | 3.08 | 0.47 | 3.05 | 0.39 |
| ReMI: time management (T1) | 0.41 | 0.25 | 0.43 | 0.31 | 0.40 | 0.28 |
| ReMI: time management (T2) | 0.48 | 0.35 | 0.75 | 0.35 | 0.79 | 0.32 |
| ReMI: time management (T3) | 0.51 | 0.34 | 0.77 | 0.38 | 0.98 | 0.37 |
| Academic performance | 19.43 | 4.12 | 21.38 | 5.12 | 22.37 | 4.55 |

Note. The control variables sex, age, trait procrastination, conscientiousness, and Time Management Questionnaire (TMQ): time management were collected at the beginning of the semester. Resource-Management Inventory (ReMI): time management was collected at the beginning (T1), mid (T2), and end of the semester (T3). Academic performance was collected at the end of the semester.

*3.2. Effectiveneess of Time Management Interventions*

We used planned contrast analyses to test the group differences expected in our hypotheses, as this is the recommended method for addressing hypotheses or specific research questions in experimental designs that involve more than two conditions, according to the American Psychology Association Guidelines for the use of statistical methods in psychology journals [95] (see also [96,97]). An alpha level of 0.05 was used for all statistical analyses.

Hypothesis 1 expected that both time management intervention groups would foster the improvement of participants' time management skills over time from the beginning to the end of the semester. To test this hypothesis, we contrasted the time management improvements in both time management intervention groups over time with those of the control group, with time management (ReMI) as a dependent variable (within-subject-factor: beginning vs. end of the semester) and group membership as a factor (contrast weights: $-2$ for the control group, 1 for each of the time management intervention groups). The contrast analysis revealed a statistically significant effect, $t(85) = 3.82$, $p < 0.001$ (one-tailed), $d = 0.83$, indicating that time management skills in both time management intervention groups improved significantly stronger over time than in the control group. Given these results, we further tested whether improvements over time also differed between the two time management intervention groups (contrast weights: 0 for the control group, $-1$ for the time management knowledge group, 1 for the time management practice group). The contrast analysis revealed a statistically significant effect, $t(85) = 2.29$, $p = 0.012$ (one-tailed), $d = 0.50$, indicating that time management improvements in the time management practice group over time were significantly stronger than in the time management knowledge group.

For exploratory purposes, we also analyzed whether the above-mentioned effects could be found already after the half of the intervention period (i.e., improvements from the beginning to the middle of the semester). The contrast that compared the two time management intervention groups with the control group revealed a statistically significant effect, $t(85) = 2.80$, $p = 0.003$ (one-tailed), $d = 0.61$, indicating that time management skills in both time management intervention groups were already significantly improved by the middle of the semester compared to the control group. By contrast, the two time management intervention groups did not differ to a significant extent at this point in time, $t(85) = 0.56$, $p = 0.289$ (one-tailed), $d = 0.12$. Figure 2 shows the development of time management skills for all intervention groups over the course of the semester.

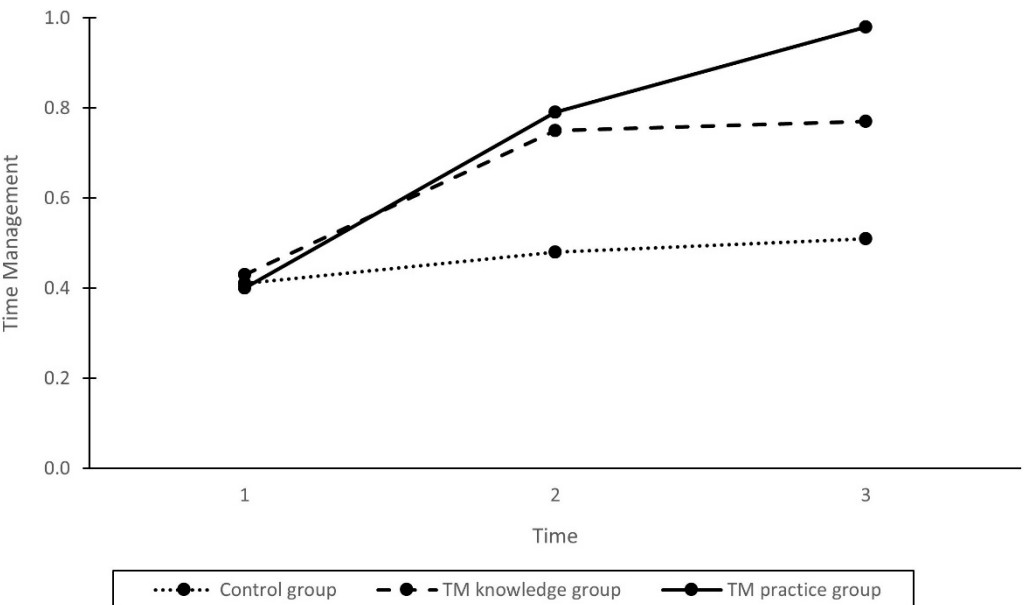

**Figure 2.** Time management (ReMI) of the intervention groups over the course of the semester (T1 = beginning, T2 = mid, T3 = end of the semester).

### 3.3. Academic Performance

Hypothesis 2 expected that the intervention that focused on time management practice, but not the intervention that focused on time management knowledge, would foster participants' academic performance at the end of the semester. To test this hypothesis, we contrasted the time management practice group with the two other groups, with exam scores as a dependent variable and group membership as a factor (contrast weights: $-1$ for the control group, $-1$ for the time management knowledge group, 2 for the time management practice group). The contrast revealed a statistically significant effect, $t(85) = 1.84$, $p = 0.035$ (one-tailed), $d = 0.41$, indicating that the time management practice group outperformed both the time management knowledge group and the control group with regard to academic performance at the end of the semester.

For exploratory purposes, we additionally contrasted each of the time management intervention groups separately with the control group. The contrast analyses revealed a statistically significant effect for the contrast between the time management practice group and the control group, $t(85) = 2.27$, $p = 0.013$ (one-tailed), $d = 0.50$, but not for the contrast between the time management knowledge group and the control group, $t(85) = 1.52$, $p = 0.066$ (one-tailed), $d = 0.34$. We also explored whether the two time management intervention groups differed in their academic performance. The contrast revealed no statistically significant effect, $t(85) = 0.84$, $p = 0.203$ (one-tailed), $d = 0.18$, indicating that the two time management intervention groups did not differ significantly with regard to their academic performance at the end of the semester.

In sum, we found that improvements in participants' time management skills over the semester were stronger in both time management intervention groups as compared to the control group, and that the time management practice group appeared superior in this regard. Since a similar pattern, albeit to a lesser extent, was found for academic performance, it could be possible that the superiority of the participants in the time management practice group over the time management knowledge and control group regarding academic performance was mediated through their superior time management skills. To test this mediation hypothesis, we conducted a mediation analysis using the SPSS macro PROCESS v4.0 [98] and calculated the 95% bootstrap percentile confidence interval of the potential indirect effect of intervention condition on academic performance via time management from 10,000 samples. The analysis revealed a statistically significant indirect effect of a $\times$ b = 1.82 [0.66, 3.46]. This finding indicated that due to the mediating function

of the time management skills, the participants in the time management practice group had an advantage of 1.82 points on the exam at the end of the semester. The path results of the mediation analysis are shown in Figure 3.

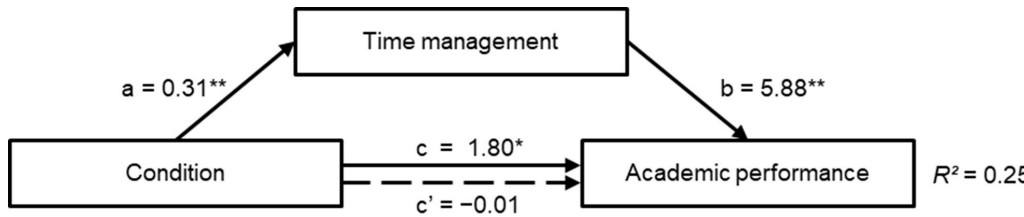

**Figure 3.** Unstandardized regression coefficients for the relationship between intervention condition (condition: 0 = control and knowledge group, 1 = practice group) and academic performance as mediated by time management (ReMI) at the end of the semester. *$p < 0.05$, **$p < 0.001$.

## 4. Discussion

We conducted two different time management interventions with freshman students who had shown a time management deficit at the beginning of their first university semester, and compared the effects on their time management skills and academic performance to a control group. In our first hypothesis, we expected that the time management skills of participants in both time management intervention groups would be significantly improved after the interventions. This assumption was supported by our results, which showed that time management skills in both intervention groups improved significantly more strongly over the semester compared to the control group. Furthermore, participants in the intervention group with the focus on time management practice showed significantly stronger time management improvements than did participants in the intervention group with the focus on time management knowledge, suggesting an advantage of time management practice in improving freshman students' time management skills. Interestingly, the time management scores of the time management practice group at the end of the semester ($M = 0.98$, $SD = 0.37$) were even close to those of the untrained participants who had shown no initial time management deficit ($M = 1.02$, $SD = 0.52$), suggesting that participants in this group were able to largely compensate their initial time management deficit over the course of the semester.

In our second hypothesis, we expected that participants in the time management practice group would show better academic performance at the end of the semester than would participants in the time management knowledge group and in the control group. Our results showed that the time management practice group indeed outperformed the time management knowledge group and the control group, but the evidence was inconsistent, as the difference between the time management practice group and the time management knowledge group was not significant. This finding was unexpected and indicates that, at least at the descriptive level, there was a small benefit in the time management knowledge group as well, and that providing time management knowledge may in itself have a positive impact on academic performance, albeit to a lesser extent.

Overall, our results support the effectiveness of time management interventions in improving freshman students' time management skills and academic performance. While both of the time management interventions led to improvements in students' time management skills, the intervention with a focus on time management practice turned out to be superior in the long term, with regard both to students' time management skills and academic performance.

### 4.1. Development of Students' Time Management over Time

Our results offer interesting insights into the development of participants' time management over time in the different intervention groups. There was a small development of time management skills in the control group. However, the mean time management improvement in this group over the course of the semester was marginal and can most

likely be attributed to maturation effects through initial experiences of university life over the time of data collection. As described before, there were significant time management improvements in both time management intervention groups over the course of the semester. However, there was a difference in the course of these improvements. Participants in both time management intervention groups first showed strong improvements in their time management skills after the initial time management training session. However, only the time management practice group also showed noticeable improvements beyond the middle of the semester, whereas the time management knowledge group did not (see Figure 2).

These findings indicate that instructing participants in time management knowledge improved their time management skills initially, but that they did not further benefit from the intervention beyond this initial improvement. The effect found in this group thus can mainly be attributed to the initial time management introduction, whereas the weekly exercises aiming to deepen and consolidate students' time management knowledge did not appear to make a substantial contribution to students' time management skills afterwards. Accordingly, contrary to earlier findings from time management interventions where imparting time management knowledge based on one or two intervention sessions [21,77] or solely based on self-directed training packages [76] led to no or only marginal improvements in time management skills, providing time management knowledge might actually be a reasonable method to improve students' time management within a comparably short period of time. Häfner and Stock [22] combined these approaches by supplementing the intervention session with additional materials that participants could be used to consolidate and improve their time management skills afterwards, which led to significant improvements in participants' self-reported use of time management behaviors. For our time management knowledge intervention, we extended this elaborate approach by directly integrating the corresponding exercises into the online-learning environment which students regularly used to learn for their course exam. This not only allowed us to distribute the intervention content over regular exercises, but also to control whether students actually completed the exercises. However, given the lack of time management improvements after the initial increase, the effects of the time management knowledge intervention appear limited and seem to not reach a level that actually enables students to further optimize their time management in a self-regulated manner. On the other hand, the ongoing time management improvements of participants observed in the time management practice group show that regular practice can help students to further improve their time management, indicating that providing guidance for students in using time management strategies and sufficient time with repeated training sessions to practice and to reflect on their performance might be necessary to help them develop their time management behaviors in a self-regulated manner.

*4.2. Time Management and the Self-Regulated Learning Process*

Participants in the time management knowledge group showed early improvements in their time management skills, but this development quickly reached a plateau (see Figure 2). On the other hand, participants in the time management practice group who were guided to use time management strategies actively and reflect on their strategy use continuously improved over the course of our data collection, although their weekly exercise stayed the same. Given the lack of time management improvements after the initial increase, the effects of the time management knowledge intervention appeared insufficient to enable students to further optimize their time management in a self-regulated manner. This appears plausible in view of studies showing that students cannot necessarily transfer their strategy knowledge into strategy use [26–28]. Process models of self-regulated learning suggest a cyclic series of forethought, performance, and self-reflection processes that learners can execute to improve their learning outcomes [14]. If improvements in students' time management knowledge *(forethought phase)* did not lead to improvements in their time management strategy use *(performance phase)*, this cycle would be interrupted, preventing students from further developing time management skills via self-reflection

on their strategy use in order to adapt their time management knowledge *(self-reflection phase)*. It can be assumed that the found plateau of time management improvements in the time management knowledge group represents exactly this kind of transfer problem. The ongoing improvements in the time management practice group, on the other hand, indicate that students in this group were able to complete this process to the point where they could profit from self-reflections on their strategy use, and to adapt their strategy knowledge for following learning cycles, which they could repeat henceforth in a self-regulated manner. At this point, it seems helpful to take a look again at the scoring of the ReMI on which our assessment of time management skills was based. Here, students' self-assessed confidence in strategy use was evaluated against the previous knowledge test for each item, so that noticeable improvements in students' time management skills could only result if they also improved their underlying strategy knowledge [83]. Consequently, despite time management knowledge not being an explicit part of their weekly exercises, students in the time management practice group further improved their time management knowledge and confidence in using the respective time management strategies until the end of our data collection, whereas in the time management knowledge group, both knowledge and confidence reached a plateau at mid-data collection. Given the absence of exercises on time management knowledge in the time management practice group after the initial introduction to time management strategies, these ongoing improvements in students' time management knowledge can be considered as an indication of successful self-regulation cycles [14], providing strong support for the consideration of time management within the theoretical framework of self-regulated learning [30]. In sum, providing basic time management knowledge and focusing on guiding students in the active use of and reflection on time management strategies seemed to pave the way for a self-regulated development of time management skills. In respect of the design of effective time management interventions, this suggests a focus on conditional (when to use a strategy) and procedural (how to use a strategy) strategy knowledge [99], and especially on practice in the use of time management strategies, over the mere imparting of declarative knowledge about time management strategies. It also highlights the relevance of implementing instructional support to promote students' strategy use [81,100–102].

### 4.3. Time Management and Academic Performance

Although the positive relationship between students' time management and academic performance is well documented [15,51,53], the reported effects of time management interventions on academic performance are quite mixed [15,16,78]. The positive relationship between students' time management skills and academic performance was supported by our results showing that time management was not only significantly correlated to academic performance, but also significantly mediated the effect of the time management practice intervention on academic performance. Accordingly, participants in the time management practice group showed better exam scores at the end of the semester, indicating an advantage of practice-based time management interventions in promoting students' academic performance, especially in the long term. Since the central element of our time management practice intervention was weekly planning and reflection on the respective plans, and also on prior plan compliance, our results also support previous findings suggesting that planning is a particularly relevant time management behavior in relation to academic performance [16].

The time management practice group showed better academic performance than the other two groups. However, the results were inconsistent in that the difference between the time management knowledge group and the time management practice group was not as clear as expected, indicating that providing time management knowledge may in itself have already had a positive impact on academic performance, albeit to a lesser extent. This was rather surprising as previous evidence suggests that improving students' time management knowledge alone should not enable them to successfully transfer this knowledge into the use of time management strategies deemed necessary to influence performance [14,27,28].

Our results showing that the time management improvements in this group stopped after an initial increase following their introduction to time management strategies can be seen as support for this transfer problem. However, we have no data to evaluate students' actual use of the time management strategies taught, and therefore cannot conclusively assess whether the improvements in strategy knowledge in this group might have led to undetected changes in their time management behaviors.

*4.4. Limitations and Furthere Research*

Previous studies providing self-directed training packages to improve students' time management found no significant improvements [77]. This might be a limitation for the effects of our intervention based on time management knowledge, since the mode of instruction here changed from a lecture for the initial time management introduction to self-directed materials for the rest of the semester. It is possible that students simply did not use the provided materials, leading to the plateau in time management improvements found after the second measure. However, we checked participants' weekly submissions regularly for all groups, and apart from minor delays, there were no problems with missing submissions. Moreover, the exercises showing effects in the time management practice group were self-directed, and of similar scope. Thus, instruction mode appears rather unlikely as an alternative explanation for the different effects. Nevertheless, it would be very interesting for future research to supplement the data collection with process data on students' learning behavior, which allows a more precise evaluation of the extent to which participants actually use the strategies taught in the regular exercises [103,104]. This would also provide further information as to whether students might be able to profit from the provided strategy knowledge to improve their time management skills in a way that is not captured by time management inventories.

Given the observed tendency of participants' time management skills in the practice group (but not the knowledge group) to improve further over the course of the semester, regardless of changes in the intervention program, it can be expected that the difference between the two time management intervention groups would increase further over time. Consequently, making longer observations of the effect of time management practice on students' time management and performance would be a very interesting approach for future research. Our data collection ended with participants' first exam phase, which is certainly an important time to have developed effective learning routines. However, this first exam phase should also offer a particularly important occasion for students to self-reflect on their learning behavior, so that researching developments in time management in the period after these exams and the corresponding results, would be particularly interesting.

Finally, our sample size limits the statistical power of the reported results. Notwithstanding the challenge of getting so many students to participate over the course of a whole semester, additional research with larger sample sizes would be essential to confirm our findings. Interestingly, there was a comparably high dropout in the control group (*N* = 7) over the course of our data collection, whereas only two participants dropped out across both time management intervention groups. Since the groups were initially comparable with regard to established demographic and performance-related variables, it is possible that participating in a time management intervention might already have influenced students' retention. Previous research has demonstrated a positive correlation between students' time management and different motivational factors [37,39,83,105–108]. It would therefore be conceivable that improvements in students' time management partly led to better performance and retention through increased motivation. For future studies, adding motivational scales to longitudinal data collections might give interesting further insights on how time management interventions affect students' academic performance.

**5. Conclusions and Implications**

Our results showed that both the intervention that focused on time management strategy knowledge and the intervention that focused on time management practice led to

significant improvements in students' time management skills. The intervention focused on time management practice appeared superior in fostering students' academic performance, but the intervention focused on time management knowledge also turned out to be beneficial in this regard, albeit to a lesser extent. Overall, the findings demonstrated that an intervention focused on time management practice was superior in improving students' time management skills compared to an intervention focused on time management knowledge, and that these improvements were reflected by students' academic performance. We could show that providing students with detailed time management knowledge did lead to time management improvements over a short period of time, but did not enable them to further improve their time management skills in a self-regulated manner. Therefore, although it might seem like a time-efficient way to improve students' time management skills, for example, as a part of more general freshman courses, imparting time management knowledge alone appears not sufficient to help students to improve their academic performance in the long term. From the perspective of the self-regulated learning framework, this means that even if students are provided with sufficient knowledge about time management strategies to master the forethought phase, instructional support is required as long as they cannot transfer their strategy knowledge into effective strategy use on their own to master the performance phase as well. On the other hand, the brief introduction to time management strategies that was part of our time management practice intervention seemed sufficient to enable students to use and practice the corresponding time management strategies, which led to ongoing improvements over the course of the semester. This supports the assumption that strategy knowledge is less of a problem for students, and that instead of further introductions to time management strategies, they need guidance on the use of the corresponding strategies and on the self-evaluation of execution quality. By supporting them to complete the whole cycle of self-regulated learning phases themselves, they are enabled to further optimize their time management behaviors throughout future learning cycles. In sum, our findings emphasize that time management interventions should primarily aim at fostering students' confidence in strategy use, and make them actively practice and reflect on their use of time management strategies over time, to develop effective self-regulated learning routines.

**Author Contributions:** S.T. conducted the study and composed the manuscript; J.W. (Julia Waldeyer), J.F. and J.W. (Joachim Wirth) supported the analyses and wrote parts of the manuscript; J.R. and D.L. provided counsel and revised the manuscript. All authors have read and agreed to the published version of the manuscript.

**Funding:** The study was funded by the German Research Foundation DFG (WI 2663/6-1).

**Institutional Review Board Statement:** Not applicable.

**Informed Consent Statement:** Informed consent was obtained from all subjects involved in the study.

**Data Availability Statement:** The data presented in this study are available from the corresponding author upon reasonable request.

**Acknowledgments:** We acknowledge support by the Open Access Publication Funds of the Ruhr-Universität Bochum. Furthermore, we would like to thank the research assistants Antonia Lücke and Charlotte Scholl for their support during the data collection for this study.

**Conflicts of Interest:** The authors declare no conflict of interest.

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
