# Peer review of "How Did It Get So Late So Soon? The Effects of Time Management Knowledge and Practice on Students’ Time Management Skills and Academic Performance"

_sustainability, doi:10.3390/su14095097_

Round 1

Reviewer 1 Report

Overall, I really appreciated learning a little bit more about time-management knowledge and practice in your context. This is a promising field and a great deal of innovative research is currently being conducted. This paper could potentially be very interesting and raises an issue that will undoubtedly be of interest to a wider readership. I invite the authors to consider the following points: improve the discussion section by more clearly providing references (relate them) to previous research. This would make it easier to “sell” the very interesting findings and explain why they are important.  Second, would it be possible to visualize methods (flowchart, etc.), as this section is quite dense? It would make it easier to digest for readers. Third, could there be a summary of the findings in relation to H1 and H2 more directly in the conclusion.

Moreover, I feel that the manuscript could be improved by a more traditional approach to organizing the content. I suggest the following headings should be added to existing ones: literature, review, implications, and further research.

Also, would it be possible to format the paper, so words are not cut in the middle at end of a sentence? For example, line 12 (similar throughout the manuscript). This would make it easier to read.

This paper could potentially be very interesting and raises an issue that will undoubtedly be of interest to a wider readership.

Author Response

Thank you very much for your comments on our research article. The comments are very insightful and helped us to further improve the quality of our manuscript, especially the methods and discussion sections.

Following your comments, we added a figure to visualize the procedure of the study, which we think noticeably improved the methods section. We have also taken up now central previous studies in the discussion again to provide clearer references to central aspects of the introduction. We added a short summary of findings regarding H1 and H2 in the conclusion as suggested, and also made references to the hypotheses more clear at the beginning of the discussion. We partially adapted our headings in the discussion section as suggested to "conclusions and implications" and "limitations and further research". However, adding the sub-headings "literature" and "review" would not fit optimally to the structure of the section since we discussed the results of the two related hypotheses in separate sections and thus would need these additional headings several times. Finally, we see your point about the splitting of words at the middle and end of some sentences. We are open to this change, but this format setting is a default in the Sustainability template and we think it is up to the editor to decide about this issue.

For the sake of clarity, we have attached a table that summarizes all suggestions and our corresponding considerations.

Reviewer 2 Report

I am not quite familiar with the contrast analyses you performed and it would be helpful to explain why the weights are -2 for the control group in the first analysis and -1 in the second.

Author Response

Thank you very much for your comments on our research article.

The contrast weights define how the groups are contrasted in the analysis, with the contrast weights always summing to 0. The description in the text shows how the groups are weighted. In our opinion, it would go a bit too far to include an excursus on contrast analyses in the research article, but following your suggestion we have now added a corresponding reference for interested readers in the text (Furr & Rosenthal, 2003; Rosnow, Rosenthal & Rubin, 2000) / “(see also [96,97])”.

For the sake of clarity, we have attached a table that summarizes all suggestions and our corresponding considerations.

Reviewer 3 Report

The article presents a topic of interest for the field of university education such as time management. Certainly, it begins with an introduction that, although it would be advisable to introduce some more current and recent reference, is solid and well built. The design of the research, despite having well-defined parts, it would be recommended that the process of ideation of the phases be made a little more explicit, in order to clarify for the reader the coherence and suitability of the phases and of the study itself as such. The results and their discussion are pertinent, and elements of judgment are obtained to be able to contribute to the scientific community as a whole. Despite this, the conclusions are a little more scarce, so it would be advisable to expand them a little more, showing greater depth and forcefulness in them.

Author Response

Thank you very much for your comments on our research article. The comments are very insightful and helped us to further improve the quality of our manuscript, especially the discussion section.

Following your comments, we expanded the discussion of the time-management interventions by clearer references to relevant literature, and the conclusions by adding more explicit practical implications. Please note that these should be interpreted very cautiously since our study is the first in that field directly comparing these types of time-management interventions for freshman students, and therefore we call for future research to replicate our results in order to finally ensure valid recommendations for practice. Furthermore, we added the self-regulated learning phases more explicitly to the discussion of time management within the self-regulated learning framework and explicitly addressed them again in the conclusions now. 

For the sake of clarity, we have attached a table that summarizes all suggestions and our corresponding considerations.
